# Effect of fascial closure using barbed sutures on incisional hernias in midline laparotomy for gynecological diseases: A multicenter randomized controlled trial (KGOG 4001)

Yong Jae Lee[1‡], Nam Kyeong Kim[2‡¤], Kidong Kim[2]*, Chel Hun Choi[3], Keun Ho Lee[4], Jong-Min Lee[5], Kwang Beom Lee[6], Dong Hoon Suh[2], Sunghoon Kim[1], Min Kyu Kim[7], Seok Ju Seong[8], Myong Cheol Lim[9]

1 Department of Obstetrics and Gynecology, Institute of Women's Life Medical Science, Yonsei University College of Medicine, Seoul, Korea, 2 Department of Obstetrics and Gynecology, Seoul National University Bundang Hospital, Seongnam, Korea, 3 Department of Obstetrics and Gynecology, Sungkyunkwan University School of Medicine, Samsung Seoul Hospital, Seoul, Korea, 4 Department of Obstetrics and Gynecology, Seoul St. Mary's Hospital, The Catholic university of Korea, Seoul, Korea, 5 Department of Obstetrics and Gynecology, Kyung Hee University School of Medicine, Kyung Hee University Hospital at Gangdong, Seoul, Korea, 6 Department of Obstetrics and Gynecology, Gachon University Gil Medical Center, Gachon University College of Medicine, Incheon, Korea, 7 Department of Obstetrics and Gynecology, Samsung Changwon Hospital, Sungkyunkwan University School of Medicine, Changwon, Korea, 8 Department of Obstetrics and Gynecology, CHA Gangnam Medical Center, CHA University School of Medicine, Seoul, Korea, 9 Center for Gynecologic Cancer, National Cancer Center, Goyang, Korea

‡ These authos are co-first authors.
¤ Current address: Department of Obstetrics and Gynecology, Korea University Ansan Hospital, Ansan, Korea
* kidong.kim.md@gmail.com

## Abstract

### Objective

To identify the effect of fascial closure using barbed sutures on the incidence of incisional hernia in patients undergoing elective midline laparotomy for gynecological diseases.

### Methods

In this multicenter, non-blind randomized controlled trial conducted from February to December 2021, patients with a BMI < 35 kg/m$^2$ and aged >18 years, scheduled for midline laparotomy, were randomly assigned to receive either barbed (experimental) or non-barbed sutures (control) for fascial closure. The primary outcome was the cumulative incidence rate of incisional hernia up to 1-year post-surgery. Secondary outcomes included incisional hernia up to 2-years post-surgery, wound complications, and postoperative pain assessed by Brief Pain Inventory-Korean scores, and Numeric Rating Scale.

which permits unrestricted use, distribution, and reproduction in any medium, provided the original author and source are credited.

**Data availability statement:** The data cannot be shared publicly due to the Privacy law. The data are available from the Data Review Board of Seoul National University Bundang Hospital (contact via drb@snubh.org) for researchers who meet the criteria for access to confidential data.

**Funding:** This work was supported by an Investigator-Initiated Study Grant from Ethicon, Inc- Johnson and Johnson MedTech (grant number WC-2019-14), which was not involved in the study design, collection, analysis, interpretation of data, writing of the report, and the decision to submit the article for publication. Kidong Kim reports grants from Ethicon, Inc- Johnson and Johnson MedTech during the study period. http://www.jnjmed.co.kr.

**Competing interests:** Kidong Kim reports grants from Ethicon, Inc- Johnson and Johnson MedTech during the study period. All other authors have no conflicts of interest to disclose.

## Results

Out of 174 patients (experimental, 86; control, 88), 36 were excluded due to dropout or loss to follow-up, leaving 138 patients (experimental, 67; control, 71) included in the analysis. The groups were balanced in terms of cancer surgeries, mean wound length, and mean surgery time. The cumulative incidence rates of incisional hernia up to 1-year (0.0% vs. 1.4%; $p > 0.999$) and 2-years (0.0% vs. 3.4%, $p = 0.496$) post-surgery did not differ significantly between the experimental and control groups. Additionally, no significant differences were observed in the incidence of wound dehiscence 4 weeks post-surgery, cumulative incidences of wound dehiscence and wound infection up to 4 weeks post-surgery, or postoperative pain scores between the groups.

## Conclusions

Fascial closure using barbed sutures resulted in no cases of incisional hernia up to 2-years post-surgery, but did not demonstrate a significant reduction in incisional hernia rates compared with the non-barbed suture.

## Trial registration

ClinicalTrials.gov NCT04643197

## Introduction

Incisional hernia, defined as any defect in the abdominal wall at a postoperative scar site, with or without a bulge, that can be identified through physical examination or imaging, is one of complications following laparotomy [1]. The incidence of incisional hernia varies widely, ranging from 10% to 23%, with risk factors including patient factors such as older age, obesity, metabolic syndrome, chronic obstructive pulmonary disease (COPD), malnutrition, and abdominal aortic aneurysm; and surgical factors including the type and size of the incision, method of abdominal wall closure, and surgical-site infections (SSI) [1,2]. The presence of an incisional hernia can have negative impacts, such as decreased quality of life, increased healthcare costs, and pain, and can lead to serious complications, such as hernia incarceration or strangulation [3,4].

Despite advancements in minimally invasive surgery, laparotomy remains an important surgical approach for gynecological cancers, especially for ovarian cancer staging surgery or radical hysterectomy for cervical cancer [5,6]. The midline incision, commonly used in these surgeries, has the highest rate of incisional hernias among the various incisional types [7]. Based on several randomized controlled trials (RCTs) and meta-analyses, recent guidelines from the European and American Hernia Societies recommend using a continuous small-bites suturing technique with a slowly absorbable suture for the closure of elective midline incisions [7]. However, even with appropriate sutures and techniques, more than 10% of patients experience incisional

hernias after abdominal surgery [8]. Therefore, there is a need to improve abdominal fascia closure methods to reduce the incidence of incisional hernias.

Barbed sutures, which are knotless surgical sutures featuring barbs on their surfaces, are widely used in various gynecological procedures, including myomectomy and vaginal cuff closure after hysterectomy [9]. Barbed sutures are as effective as, or even superior to, conventional sutures in many situations, including the uterine closure in laparoscopic myomectomies, vaginal cuff closure in total laparoscopic hysterectomies, and robotic sacrocolpopexy. The STRATAFIX Symmetric PDS Plus, a type of barbed suture, has higher tensile strength and wound-holding strength than conventional sutures; therefore, it can be used to suture high-tension areas such as the fascia [10]. Despite these potential benefits, research on the use of barbed sutures for abdominal fascial closure in gynecological midline laparotomies is lacking.

The present study aims to identify the effect of fascial closure using barbed sutures on the incidence of subsequent incisional hernia in patients who underwent midline laparotomy for gynecological diseases. We hypothesized that fascial closure using barbed sutures would reduce the cumulative incidence of incisional hernia compared to the use of non-barbed sutures.

## Materials and methods

### Study design

The current study was the Korean Gynecologic Oncology Group (KGOG) 4001 study, a prospective, multicenter, non-blind RCT conducted at nine institutes in Korea. The study was approved by the institutional review board of all participating hospitals (Seoul National University Bundang Hospital, B-2005-615-004; Samsung Seoul Hospital, 2020-10-044; Institute of Women's Life Medical Science, 1-2020-0080; Seoul St. Mary's Hospital, KC20DIDI0967; Kyung Hee University Hospital at Gangdong, KHNMC 2020-10-027; Gachon University Gil Medical Center, GCIRB2020-453; Samsung Changwon Hospital, 2020-10-002; CHA Gangnam Medical Center; 2020-11-001; National Cancer Center, NCC2021-0132) and was conducted according to the Declaration of Helsinki. The trial was registered at ClinicalTrials.gov (NCT04643197) before the inclusion of the first participant (Date of registration: 25/11/2020; Date of initial participant enrollment: 19/03/2021). Participant recruitment began on 01/03/2021 at eight participating institutions. Due to a delay in initial IRB approval, recruitment at the National Cancer Center started later, on 21/04/2021. Recruitment at all sites was completed by 08/11/2021. The follow-up period for the primary outcome was from May to December 2022. All participants signed an informed consent form after receiving adequate information about the study before enrollment.

Patients scheduled to undergo elective midline laparotomy for gynecological diseases, aged >18 years, and with an Eastern Cooperative Oncology Group performance status of 0–2 were eligible for the study. The exclusion criteria were as follows: (1) history of incisional or concurrent incisional hernia; (2) concurrent pregnancy; (3) history of prior radiation therapy to the abdomen or pelvis; (4) allergic to polydioxanone (PDS); (5) diseases that affect wound healing, such as uncontrolled diabetes mellitus, autoimmune vasculitis, liver cirrhosis, or coagulation disorder; (6) body mass index (BMI) > 35 kg/m$^2$; (7) previous or future use of drugs that may affect wound healing (e.g., bevacizumab) within 4 weeks before or after surgery; (8) history of midline laparotomy within 6 months; and (9) undergoing surgery due to infection. The enrolled patients were randomized in a 1:1 ratio into the experimental (with barbed suture) and control (with non-barbed suture) groups. Randomization was conducted centrally through the KGOG using a secure electronic Case Report Form (eCRF) [REDCap] system, with pre-generated randomization tables created by the Clinical Research Collaboration Center. When patients were admitted for surgery and eligibility was confirmed, each participating center requested randomization from KGOG. Block randomization with stratification by institution, BMI, and surgical indication was applied. Allocation was disclosed only immediately before fascial closure by telephone confirmation with KGOG, ensuring allocation concealment.

The data were accessed for research purposes between January 1, 2023, and January 1, 2024. The authors did not have access to identifiable personal information during or after data collection. All data were anonymized prior to analysis in accordance with institutional privacy policies. The data cannot be shared publicly due to the Privacy Law. However, the

data are available from the Data Review Board of Seoul National University Bundang Hospital for researchers who meet the criteria for access to confidential data.

## Surgical procedures

Peritoneal closure was performed before fascial suturing in all patients, except those who underwent peritonectomy. The experimental and control groups used the STRATAFIX Symmetric PDS Plus (suture size: 1/0, suture length: 45 cm, needle size: 40 mm [CT needle]; Ethicon, Somerville, NJ, USA) and PDS Plus (suture size: 1/0, suture length: 90 cm, needle size: 40 mm [CT needle]; Ethicon, Somerville, NJ, USA), respectively, for fascial suturing. Fascial sutures were performed as continuous running sutures with stitch sparing of 5–8 mm and small bites of 5–8 mm in both groups. According to the operator's judgment, several fascial interrupted sutures and additional subcutaneous sutures were allowed in both groups. Negative-pressure subcutaneous drains (BAROVAC SS100M, SEWOON MEDICAL, Korea) were inserted according to the results of subcutaneous drain randomization, which was conducted independently of the fascial suturing randomization. The skin was closed using staples or non-absorbable sutures.

## Outcomes

The primary outcome was the cumulative incidence rate of incisional hernia up to 1 year post-surgery according to fascial closure through the randomization of suture materials. The investigators conducted a physical examination as the primary method to diagnose incisional hernia during outpatient follow-up visits, and if necessary, abdominal ultrasound or computed tomography (CT) scans were performed. Additionally, incidental findings of incisional hernia on imaging studies performed during the follow-up period were also included in the diagnosis. Secondary outcomes were the cumulative incidence rate of incisional hernia up to 2 years post-surgery; cumulative incidence rate of wound infection up to 4 weeks post-surgery; cumulative incidence rate of wound dehiscence up to 4 weeks post-surgery; incidence rate of wound dehiscence 4 weeks post-surgery; Brief Pain Inventory-Korean (BPI-K) score at baseline, postoperative day 2 and postoperative day 4; Numeric Rating Scale (NRS) score from surgery to postoperative day 4, and other reported treatment-related adverse events. The presence of incisional hernia up to 2 years post-surgery was additionally assessed through a retrospective review of medical records.

## Statistical analyses

Based on previous studies, the 1-year cumulative incidence of incisional hernias in the control group was estimated to be 0.13 [8]. No existing studies have investigated the incidence of incisional hernia after the use of barbed sutures in abdominal fascial closure. However, considering that some studies utilizing conventional suture methods have reported a very low incidence of incisional hernia (1–2%), we estimated the 1-year cumulative incidence of incisional hernia in the experimental group to be 0.03. To determine the appropriate sample size, we set the one-sided alpha level at 0.05 and the power at 0.8, accounting for a dropout rate of 0.06. The calculated sample sizes for each group were 87. Therefore, the total target enrollment for the study was set at 174 participants.

The modified intention-to-treat (mITT) population was defined as all randomized patients who underwent surgery and had evaluable data for incisional hernia up to 1 year post-surgery. The per-protocol (PP) population was defined as patients who adhered to the assigned intervention without major protocol deviations, completed scheduled follow-up, and had evaluable outcome data.

Pearson's $\chi^2$ or Fisher's exact test was used to compare categorical variables between the experimental and control groups, and Student's t-test or Mann–Whitney U test was used to analyze continuous variables. All statistical analyses were performed using the Statistical Package for the Social Sciences for Windows (version 27.0; IBM Corp., Armonk, NY). Statistical significance was set at $p < 0.05$.

## Results

A total of 174 patients were randomized, with 86 allocated to the experimental group and 88 to the control group. Nine patients dropped out before or during surgery (experimental, 5; control, 4). An additional 27 patients (experimental, 14; control, 13) were lost to follow-up at 1-year post-surgery, with no cases of incisional hernia observed until the last follow-up. The remaining 138 patients (experimental, 67; control, 71) were included in the final analysis. As all patients underwent fascial closure as assigned, the mITT analysis was identical to the PP analysis. A flowchart of participation in the study is shown in Fig 1.

Baseline characteristics of the study population are summarized in Table 1. Ten patients (7.2%) had a BMI > 30 kg/m$^2$. Additionally, 24 (17.4%) patients had a history of at least one midline laparotomy. The mean wound length was 24.7 cm, and the mean surgery time was 231.7 minutes. Pathological diagnoses after surgery revealed that 25 patients (18.1%) had benign conditions, whereas 113 (81.9%) were diagnosed with malignancies. There were no statistically significant differences in age at surgery, frequency of BMI > 30 kg/m$^2$, frequency of current smoking, history of steroid medication use, chemotherapy history within 6 months of surgery, history of bevacizumab administration within 6 months of surgery, or history of previous midline laparotomy between the experimental and control groups (Table 2).

The surgical factors and outcomes according to fascial closure are shown in Table 3. There were no significant differences in wound length between the experimental and control groups; however, the mean suture length used was significantly shorter in the experimental group than in the control group (54.8 cm vs. 71.0 cm, p = 0.001). The suture length to wound length ratio was also significantly lower in the experimental group compared to the control group (2.4 vs. 3.0, p = 0.007). Other surgical factors, including surgery time, subcutaneous drain, subcutaneous suture, skin suture, and postoperative diagnosis, were not significantly different between the two groups.

No incisional hernias were reported up to 4 weeks post-surgery in either group, and only one case of incisional hernia was reported in the control group up to 1 year post-surgery, with no significant difference between the two groups (p > 0.999). The patient was diagnosed with an incisional hernia during a physical examination at the 1-year follow-up, but remained asymptomatic and did not receive additional treatment. The cumulative incidence rate of incisional hernia up to 2 years post-surgery showed no significant difference between the experimental and control groups (0.0% [0/55] vs. 3.4% [2/58], p = 0.496). Another patient was found to have an asymptomatic incisional hernia on CT at the 2-year follow-up, but did not receive treatment. The incidence rate of wound dehiscence at 4 weeks post-surgery, cumulative incidence rate of wound dehiscence up to 4 weeks post-surgery, and cumulative incidence rate of wound infection up to 4 weeks post-surgery were not significantly different between the groups.

The total BPI-K score and individual item scores at baseline and on postoperative day 2, and day 4 were not significantly different between the experimental and control groups (S1–S3 Tables). Additionally, there was no significant difference in the NRS scores from the day of surgery to postoperative day 4 between the two groups (S4 Table). Immediate postoperative complications during hospitalization included one case of acute kidney injury, one case of deep vein thrombosis, one case of stroke on postoperative day 3, and one case of postoperative ileus, which improved with conservative treatment. No complications were reported at the 4-week or 1-year follow-ups.

## Discussion

The current study failed to demonstrate a significant improvement in the incidence of incisional hernia with fascial closure using barbed sutures in women with BMI < 35 kg/m$^2$ who underwent midline laparotomy. There were no significant differences in the incidence of wound dehiscence 4 weeks post-surgery, cumulative incidence rate of wound dehiscence, cumulative incidence rate of wound infection up to 4 weeks post-surgery, or postoperative pain scores based on the methods of fascial closure.

Several meta-analyses and RCTs have shown that slowly absorbable continuous sutures reduce the incidence of incisional hernia compared with interrupted sutures [11,12]. A 2017 Cochrane review reported that monofilament

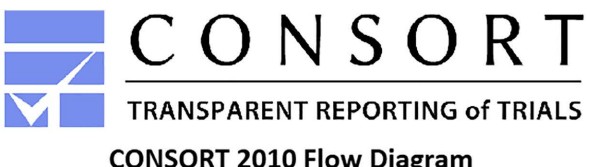

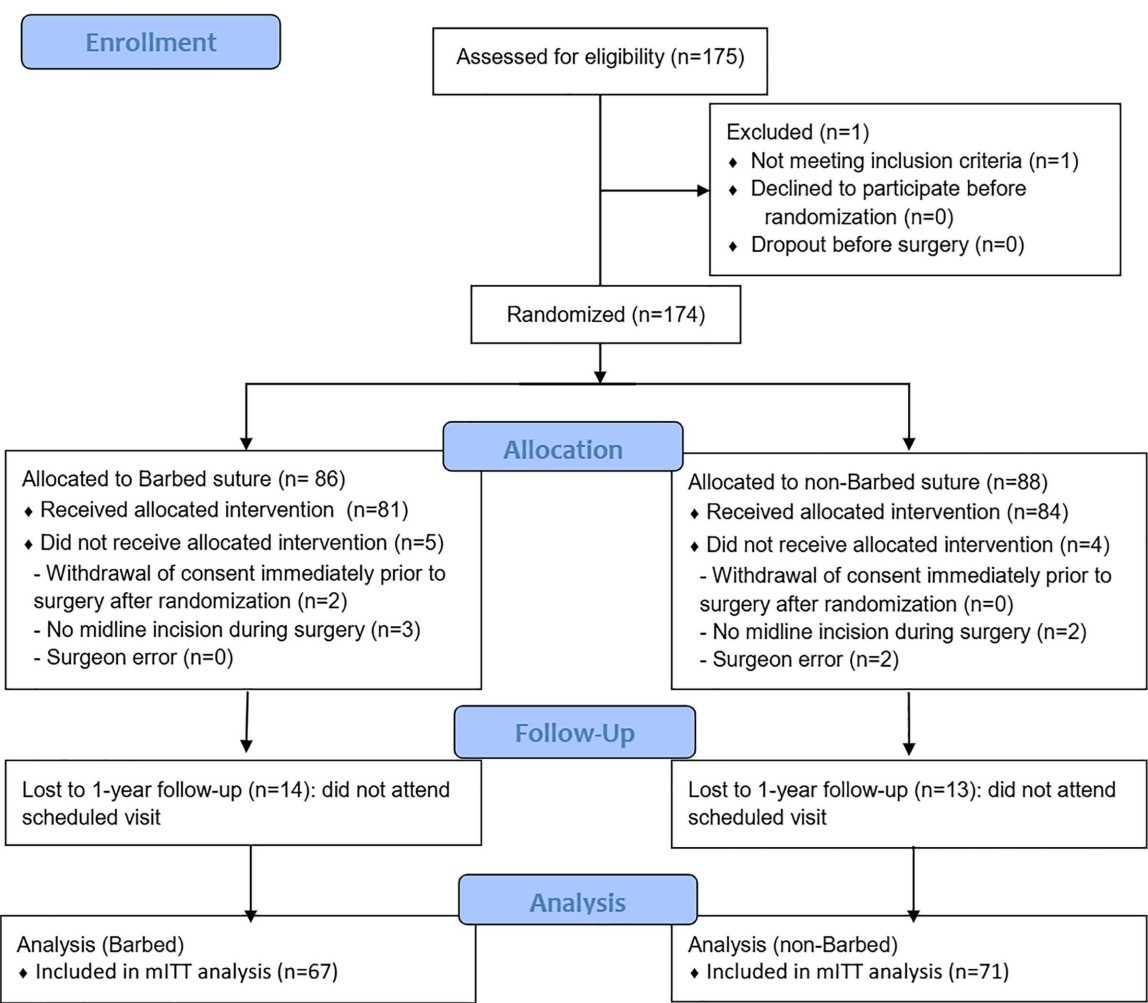

**Fig 1. CONSORT (Consolidated Standards of Reporting Trials) 2010 flow diagram.**

sutures might reduce the risk of incisional hernia, whereas absorbable sutures might reduce the risk of sinus or fistula tract formation [13]. One RCT reported a significantly lower incidence of incisional hernia at the 1-year follow-up in a small-bite group (tissue bites of 5 mm every 5 mm) than in a large-bite group (bites of 1 cm every 1 cm) (13% vs. 21%, p = 0.0220) [8]. Current guidelines recommend a continuous small-bites suturing technique with a slowly absorbable suture to close elective midline incisions, which was also followed in the control group in the current study [7].

**Table 1. Baseline characteristics of the study population (N = 138).**

| Variables | Values |
|---|---|
| Age at surgery (years) | 54.1 ± 11.2 |
| ASA score | |
| 1 | 35 (25.4) |
| 2 | 93 (67.4) |
| 3 | 10 (7.2) |
| BMI > 30 kg/m$^2$ | 10 (7.2) |
| Height (cm) | 157.3 ± 5.4 |
| Current smoking | 7 (5.1) |
| History of steroid medication | 8 (5.8) |
| The number of previous midline laparotomy | |
| 0 | 114 (82.6) |
| 1 | 18 (13.0) |
| 2 | 5 (3.6) |
| 3 | 1 (0.7) |
| History of chemotherapy[a] | 17 (12.3) |
| History of bevacizumab administration[a] | 3 (2.2) |
| Wound length (cm)[b] | 24.7 ± 7.7 |
| Additional antibiotics during operation | 5 (3.6) |
| Subcutaneous drain insertion | 65 (47.1) |
| Additional subcutaneous suture | 96 (69.6) |
| Skin suture | |
| Stapler | 115 (83.3) |
| Vertical mattress suture | 5 (3.6) |
| Subcuticular suture | 4 (2.9) |
| Skin bond | 14 (10.1) |
| Post-operative diagnosis | |
| Benign | 25 (18.1) |
| Malignancy | 113 (81.9) |
| Surgery time (minutes) | 231.7 ± 123.0 |
| Post-operative treatment | |
| No | 46 (33.3) |
| Chemotherapy | 73 (52.9) |
| Radiotherapy (including CCRT) | 18 (13.0) |
| Other | 1 (0.7) |

Values are presented as mean ± standard deviation or number (%).

[a]within 6 months of surgery.

[b]Two data were missing.

ASA, American Society of Anestheologist; BMI, body mass index; CCRT, Concurrent chemoradiotherapy.

In a porcine model study, barbed sutures were compared with non-barbed absorbable sutures for fascial repair of paramedian incisions, showing that barbed sutures provided tensile strength equivalent to traditional non-barbed sutures without adverse events such as wound dehiscence or incisional hernia [14]. This finding suggests that barbed sutures are a reliable option for fascial closure, offering similar mechanical strength without increasing the risk of complications. A single-center retrospective study on laparoscopic or robotic single-incision cholecystectomy reported that the incidence of

**Table 2. Clinical factors between experimental and control groups.**

| | Experimental (barbed suture) n = 67 | Control (non-barbed suture) n = 71 | p value |
|---|---|---|---|
| Age at surgery (years) | 54.9 ± 10.0 | 53.3 ± 12.3 | 0.422 |
| BMI > 30 kg/m² | 5 (7.5) | 5 (7.0) | 0.924 |
| Height (cm) | 156.9 ± 5.0 | 157.7 ± 5.8 | 0.371 |
| ASA score | | | 0.370 |
| 1 | 18 (26.9) | 17 (23.9) | |
| 2 | 46 (68.7) | 47 (66.2) | |
| 3 | 3 (4.5) | 7 (9.9) | |
| Current smoking | 2 (3.0) | 5 (7.0) | 0.442 |
| History of steroid medication | 3 (4.2) | 5 (7.0) | 0.719 |
| History of chemotherapy[a] | 6 (9.1) | 11 (15.5) | 0.243 |
| History of bevacizumab administration[a] | 0 (0.0) | 3 (4.2) | 0.245 |
| The number of previous midline laparotomy | | | 0.987 |
| 0 | 55 (82.1) | 59 (83.1) | |
| 1 | 9 (13.4) | 9 (12.7) | |
| 2 | 3 (4.5) | 2 (2.8) | |
| 3 | 0 (0.0) | 1 (1.4) | |
| History of previous midline laparotomy | | | 0.876 |
| Yes | 55 (82.1) | 59 (83.1) | |
| No | 12 (17.9) | 12 (16.9) | |

Values are presented as mean ± standard deviation or number (%).

[a]within 6 months of surgery.

BMI, body mass index.

incisional hernia was significantly lower in the barbed suture group than in the monofilament suture group (0.0% vs. 0.7%, p = 0.021) [15]. Similar to our study, the incidence of incisional hernias was very low. Differences in the surgical context, such as the smaller fascia opening of 3 cm, variations in surgical time, and type of surgery, make direct comparisons challenging. A previous study analyzed incisional hernia only in patients who returned with symptoms such as bulging after routine follow-up for one month after surgery, potentially underestimating the true incidence of hernia [15].

Although this RCT trial did not demonstrate statistical superiority of barbed sutures in reducing the incidence of incisional hernia compared with non-barbed sutures, it is noteworthy that no cases of incisional hernia occurred in the barbed suture group up to 2 years post-surgery. Additionally, in the barbed suture group, wound infection, wound dehiscence, and postoperative pain scores did not significantly increase compared to those in the non-barbed suture group. Therefore, fascial closure using barbed sutures can be considered a feasible method in patients with a BMI < 35 kg/m² who have undergone midline laparotomy for gynecological disease. However, because the benefits of reducing incisional hernia and wound complications were not demonstrated in this study, a large-scale RCT reflecting the low incidence of incisional hernia is needed for barbed suture to become a standard method.

We reported a lower incidence of incisional hernia in the control group using the conventional method than in previous studies. Our protocol used a small-bite suturing method that has been proven to reduce incisional hernias by 48% [8,16]. SSI is a major risk factor for the development of incisional hernias [17]. Gynecological surgical wounds are typically classified as clean-contaminated wounds with an SSI rate of < 10%, whereas colorectal surgical wounds are classified as contaminated wounds with SSI rates of 13–20% [18]. Previous RCTs focused mainly on colorectal surgeries or included only a small proportion of gynecological surgeries, which might have led to a higher incidence of incisional hernias due to

**Table 3. Surgical factors and outcomes between experimental and control groups.**

| | Experimental (barbed suture) n=67 | Control (non-barbed suture) n=71 | p value |
|---|---|---|---|
| Wound length (cm)[a] | 24.3±8.1 | 25.1±7.4 | 0.587 |
| Suture length used (cm)[a] | 54.8±21.4 | 71.0±31.1 | 0.001 |
| Suture length to wound length ratio[a] | 2.4±1.1 | 3.0±1.5 | 0.007 |
| Surgery time (minutes) | 219.4±132.1 | 243.2±113.6 | 0.257 |
| Additional antibiotics during operation | 3 (4.5) | 2 (2.8) | 0.674 |
| Additional subcutaneous suture | 45 (67.2) | 51 (71.8) | 0.552 |
| Subcutaneous drain insertion | 30 (44.8) | 35 (49.3) | 0.595 |
| Post-operative diagnosis | | | 0.088 |
| Benign | 16 (23.9) | 9 (12.7) | |
| Malignancy | 51 (76.1) | 62 (87.3) | |
| Skin suture | | | 0.735 |
| Stapler | 55 (82.1) | 60 (84.5) | |
| Vertical mattress suture | 2 (3.0) | 3 (4.2) | |
| Subcuticular suture | 4 (6.0) | 0 (0.0) | |
| Skin bond | 6 (9.0) | 8 (11.3) | |
| Wound infection up to 4 weeks post-surgery | 0 (0.0) | 2 (2.8) | 0.497 |
| Wound dehiscence up to 4 weeks post-surgery | 4 (6.0) | 8 (11.3) | 0.368 |
| Wound dehiscence 4 weeks post-surgery | 1 (1.5) | 2 (2.8) | >0.999 |
| Incisional hernia up to 4 weeks post-surgery | 0 (0.0) | 0 (0.0) | |
| Incisional hernia up to 1 year post-surgery | 0 (0.0) | 1 (1.4) | >0.999 |
| Incisional hernia up to 2 year post-surgery | 0/55 (0.0) | 2/58 (3.4) | 0.496 |

Values are presented as mean±standard deviation or number (%).

[a]Two data were missing.

increased SSI rates compared to our study, which included only gynecological surgeries [2,8]. Obesity is also known to be a risk factor for incisional hernia owing to its negative impact on wound healing; however, the criteria for obesity vary across studies [17,19]. In our study, morbidly obese patients were not included, and only 7.2% of the participants were classified as obese according to WHO criteria, with the majority having a BMI < 30 kg/m$^2$.

Currently, one RCT is ongoing to evaluate the effect of absorbable barbed sutures on incisional hernia in midline fascial closure during minimally invasive surgery for colorectal and gastric cancers with wound lengths of > 1 cm but less than 10 cm. The outcomes of this ongoing trial will be especially relevant for comparison with those of our study, given the similar focus on midline fascial closure, albeit in a different patient population, surgical setting, and wound type [20].

This is the first RCT to compare barbed and non-barbed sutures in the fascial closure of midline abdominal incisions. Additionally, the study focused on gynecological surgery patients, which allowed for a relatively homogeneous wound type across the study population. However, there are several limitations to consider. First, it was conducted using non-blinded randomization. It is highly likely that the surgeons' efforts to prevent incisional hernia were reflected in the surgical procedure. The current study permitted the use of additional fascial interrupted sutures at the discretion of the surgeon, but the proportion and length of these sutures were not documented, which might have contributed to the mean suture length to wound length ratio being less than 4. Because the surgeons were responsible for both the intervention and outcome assessment, the risk of performance and detection bias cannot be fully excluded. However, patients were not informed of their allocation, and randomization results were not documented in the medical records but stored only in the eCRF, thereby minimizing potential bias during incisional hernia diagnosis. In addition, the study did not consider all patient risk

factors that may influence the development of incisional hernias, such as metabolic syndrome, COPD, malnutrition, and abdominal aortic aneurysm. Given the lower-than-expected 1-year cumulative incidence rate of incisional hernia in the control group, along with the high dropout and loss to follow-up rates, this study did not appear to have sufficient power to demonstrate statistical superiority. This study may be subject to potential reporting bias due to perioperative dropouts and loss to follow-up, as well as performance bias related to the use of reinforced sutures, additional subcutaneous sutures, and subcutaneous drains. These limitations should be taken into consideration when interpreting the findings. To address concerns regarding missing data, we performed a sensitivity analysis under both best-case and worst-case assumptions. In the best-case scenario, where all dropouts and follow-up losses were assumed not to have developed incisional hernia, the results were identical to the observed outcomes (0 vs. 1 event). In the worst-case scenario, where all such patients were assumed to have developed incisional hernia, the incidence was 22.1% (19/86) in the barbed suture group and 20.5% (18/88) in the non-barbed suture group. Importantly, in both scenarios, the overall interpretation of the primary outcome remained unchanged, suggesting that our findings are robust despite the relatively high dropout and follow-up loss rates.

## Conclusion

Fascial closure using barbed sutures resulted in zero incisional hernia up to 2 years post-surgery in patients with a BMI $< 35$ kg/m$^2$ who underwent midline laparotomy for gynecological disease; however, we failed to demonstrate a reduction in incisional hernia compared with the non-barbed suture method. The incidences of wound dehiscence and infection, and post-operative pain scores were not significantly different based on whether barbed sutures were used for fascial closure.

## Supporting information

**S1 Table. BPI-K score at baseline between experimental and control group.**
(DOCX)

**S2 Table. BPI-K score at postoperative day 2 between experimental and control group.**
(DOCX)

**S3 Table. BPI-K score at postoperative day 4 between experimental and control group.**
(DOCX)

**S4 Table. NRS pain score from surgery to postoperative day 4 between experimental and control group.**
(DOCX)

**S5 File. Clinical trial protocol of KGOG 4001 (English version).**
(DOCX)

**S6 File. Clinical trial protocol of KGOG 4001 (original, Korean version).**
(DOCX)

**S7 File. CONSORT 2025 checklist.**
(DOCX)

## Author contributions

**Conceptualization:** Kidong Kim.

**Data curation:** Yong Jae Lee.

**Formal analysis:** Yong Jae Lee, Nam Kyeong Kim.

**Funding acquisition:** Kidong Kim.

**Investigation:** Yong Jae Lee, Nam Kyeong Kim.

**Methodology:** Yong Jae Lee, Nam Kyeong Kim, Kidong Kim.

**Project administration:** Kidong Kim.

**Resources:** Yong Jae Lee, Kidong Kim, Chel Hun Choi, Keun Ho Lee, Jong-Min Lee, Kwang Beom Lee, Dong Hoon Suh, Sunghoon Kim, Min Kyu Kim, Seok Ju Seong, Myong Cheol Lim.

**Supervision:** Kidong Kim.

**Validation:** Yong Jae Lee, Chel Hun Choi, Keun Ho Lee, Jong-Min Lee, Kwang Beom Lee, Dong Hoon Suh, Sunghoon Kim, Min Kyu Kim, Seok Ju Seong, Myong Cheol Lim.

**Visualization:** Nam Kyeong Kim.

**Writing – original draft:** Nam Kyeong Kim.

**Writing – review & editing:** Yong Jae Lee, Nam Kyeong Kim, Kidong Kim, Chel Hun Choi, Keun Ho Lee, Jong-Min Lee, Kwang Beom Lee, Dong Hoon Suh, Sunghoon Kim, Min Kyu Kim, Seok Ju Seong, Myong Cheol Lim.

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
