## [Decision Letter · Decision Letter 0]

18 Aug 2025

Thank you for submitting your manuscript to PLOS ONE. After careful consideration, we feel that it has merit but does not fully meet PLOS ONE’s publication criteria as it currently stands. Therefore, we invite you to submit a revised version of the manuscript that addresses the points raised during the review process.

We look forward to receiving your revised manuscript.

Kind regards,

Marco Clementi, Associate Professor

Academic Editor

PLOS ONE

Journal Requirements:

3. Thank you for stating the following in the Competing Interests/Financial Disclosure section:

Kidong Kim reports grants from Ethicon, Inc- Johnson and Johnson MedTech during the study period.

All other authors have no conflicts of interest to disclose.

We note that one or more of the authors have an affiliation to the commercial funders of this research study: Ethicon, Inc- Johnson and Johnson MedTech

4. Please remove all personal information, ensure that the data shared are in accordance with participant consent, and re-upload a fully anonymized data set.

Additional guidance on preparing raw data for publication can be found in our Data Policy (https://journals.plos.org/plosone/s/data-availability#loc-human-research-participant-data-and-other-sensitive-data) and in the following article: http://www.bmj.com/content/340/bmj.c181.long .

Additional Editor Comments:

Thank you for submitting your study. The study is interesting but changes must be made as per reviewers' requests.

Reviewers' comments:

Reviewer's Responses to Questions

**Comments to the Author**

1. Is the manuscript technically sound, and do the data support the conclusions?

Reviewer #1: Partly

Reviewer #2: Partly

2. Has the statistical analysis been performed appropriately and rigorously?

Reviewer #1: No

Reviewer #2: No

3. Have the authors made all data underlying the findings in their manuscript fully available?

Reviewer #1: Yes

Reviewer #2: No

4. Is the manuscript presented in an intelligible fashion and written in standard English?

Reviewer #1: Yes

Reviewer #2: Yes

Reviewer #1: In this multicenter, non-blind randomized controlled trial patients were randomly assigned to receive either barbed (experimental) or non-barbed sutures (control) for fascial closure. The primary outcome was the cumulative incidence rate and secondary outcomes were also explored. The study was reasonably designed with sample size and power considerations yielding what appears to be an adequate sample and about 80% power. The effect sizes and dropout rate were included resulting in the calculated number of subjects needed to compare the incidence incisional hernia rates between the two groups. The analysis will be parametric or non parametric depending on the normality or non normality of the data. The final analysis was appropriate for the data type.

The investigators need to justify the one sided statistical approach compared to the two sided most likely needed in this type of comparison. This may be a mute point since the results were not statistically different and will remain as such with a two sided test.

The analysis supported the conclusion that the cumulative incidence rate of incisional hernia up to 2 years post-surgery showed no significant difference between the experimental and control groups (0.0% [0/55] vs. 3.4% [2/58], p=0.496). Further results are seen in Table 3. Obviously, the investigators did not demonstrate a reduction in incisional hernia compared with the non-barbed suture method.

Reviewer #2: a. The design has high risk of bias: reporting bias as they took patients out from analysis and performance bias by letting reinforced sutures be done and randomizing also subcutaneous drains.

b. Randomization is not clear. It is mentioned made in blocks but not who did it, how and when the patient was randomized in each center.

c. About the blinding , it was mentioned that surgeons where not blinded, but it is not clear if the patient was not blinded either? Or the person in charge of assessing the outcomes, some blinding could have been done there.

d. I have my doubts about analysis. It is not mentioned the existence of a Data and Safety Monitoring Board, or not doing an interim analysis being the first RCT on this kind of suture.

There is a paragraph mentioning that intention to treat analysis was similar to per protocol analysis but the definition of ITT mentioned is not justified as there were approximately 20% of patients not reported, and not included in the analysis even though they were not only allocated but operated on, looking at the consort flowchart. So that introduces confusion about the definition used for ITT when there are so many loss of follow ups.

Which is part of a rigorous statistical analysis in a RCT. Some data is missing and not following CONSORT guideline, only the statistical methods to compare groups for primary and secondary outcomes is described, not including harms.

But it is missing:

• Definition of who is included in each analysis and in which group, ITT? Or Per protocol? how

• How did you handle missing data in the analysis, excluding them?

• Methods for any additional analysis such as sensitivity analysis, confounders?

e. It is clear that data is not publicly available but how in who can see them

f. Many CONSORT items are missing

g. Table 1 is not necessary, patients characteristics should have been presented by randomized groups lookinf for confounders

h. Small sample size. Authors recognized the study is underpowered

**Do you want your identity to be public for this peer review?** For information about this choice, including consent withdrawal, please see our Privacy Policy

Reviewer #1: No

Reviewer #2: No

---

## [Author Response · Author response to Decision Letter 1]

9 Sep 2025

We thank the Academic Editor and Reviewers for their thorough evaluation of our manuscript and for providing constructive comments that have helped us to improve the clarity and quality of the paper. Below, we provide a point-by-point response to each comment. All changes in the revised manuscript have been marked using track changes.

Reviewer #1

Comment 1:

In this multicenter, non-blind randomized controlled trial patient were randomly assigned to receive either barbed (experimental) or non-barbed sutures (control) for fascial closure. The primary outcome was the cumulative incidence rate and secondary outcomes were also explored. The study was reasonably designed with sample size and power considerations yielding what appears to be an adequate sample and about 80% power. The effect sizes and dropout rate were included resulting in the calculated number of subjects needed to compare the incidence incisional hernia rates between the two groups. The analysis will be parametric or non-parametric depending on the normality or non-normality of the data. The final analysis was appropriate for the data type. The investigators need to justify the one-sided statistical approach compared to the two sided most likely needed in this type of comparison. This may be a moot point since the results were not statistically different and will remain as such with a two-sided test.

Response 1:

We thank the reviewer for this important observation. We fully acknowledge that in comparative clinical trials, a two-sided test is generally preferred. Our initial rationale for selecting a one-sided approach was based on the hypothesis that barbed sutures would not increase the risk of incisional hernia compared with conventional sutures, but rather potentially reduce it.

Comment 2:

The analysis supported the conclusion that the cumulative incidence rate of incisional hernia up to 2 years post-surgery showed no significant difference between the experimental and control groups (0.0% [0/55] vs. 3.4% [2/58], p=0.496). Further results are seen in Table 3. Obviously, the investigators did not demonstrate a reduction in incisional hernia compared with the non-barbed suture method.

Response 2:

We agree with the reviewer’s interpretation. Our study hypothesis was that fascial closure using barbed sutures would reduce the cumulative incidence of incisional hernia compared with non-barbed sutures. In line with this, we emphasized in the Conclusion that barbed sutures did not demonstrate superiority in reducing incisional hernia rates. However, it is noteworthy that fascial closure with barbed sutures resulted in no cases of incisional hernia up to 2 years post-surgery, which suggests the feasibility and potential clinical value of this technique, despite the lack of demonstrated superiority in this trial.

Page 17, Line 283-292 (Discussion section)

Although this RCT trial did not demonstrate statistical superiority of barbed sutures in reducing the incidence of incisional hernia compared with non-barbed sutures, it is noteworthy that no cases of incisional hernia occurred in the barbed suture group up to 2 years post-surgery. Additionally, in the barbed suture group, wound infection, wound dehiscence, and postoperative pain scores did not significantly increase compared to those in the non-barbed suture group. Therefore, fascial closure using barbed sutures can be considered a feasible method in patients with a BMI < 35 kg/m² who have undergone midline laparotomy for gynecological disease. However, because the benefits of reducing incisional hernia and wound complications were not demonstrated in this study, a large-scale RCT reflecting the low incidence of incisional hernia is needed for barbed suture to become a standard method.

Reviewer #2

Comment a: The design has high risk of bias: reporting bias as they took patients out from analysis and performance bias by letting reinforced sutures be done and randomizing also subcutaneous drains.

Response a:

We thank the reviewer for this important point. Patients who dropped out before or during surgery, or who were lost to follow-up, were excluded from the analysis, and all exclusions with reasons are reported in the CONSORT flow diagram; we acknowledge this may have introduced reporting bias. Although additional subcutaneous sutures and subcutaneous drains could introduce performance bias, these factors were balanced between groups (67.2% vs. 71.8%, p = 0.552; 44.8% vs. 49.3%, p = 0.595). We have also addressed these potential biases in the Limitations section of the Discussion.

Page 19, Line 331-334

This study may be subject to potential reporting bias due to perioperative dropouts and loss to follow-up, as well as performance bias related to the use of reinforced sutures, additional subcutaneous sutures, and subcutaneous drains. These limitations should be taken into consideration when interpreting the findings.

Comment b: Randomization is not clear. It is mentioned made in blocks but not who did it, how and when the patient was randomized in each center.

Response b:

We appreciate this important comment. Randomization was conducted centrally through the KGOG using a secure eCRF (REDCap) system, with pre-generated randomization tables created by the Clinical Research Collaboration Center. When patients were admitted for surgery and eligibility was confirmed, each participating center requested randomization from KGOG. Block randomization with stratification by institution, BMI, and surgical indication was applied. Allocation was disclosed only immediately before fascial closure by telephone confirmation with KGOG, ensuring allocation concealment. We have added this clarification to the Methods section (Page 8, Line 136-142).

Comment c: About the blinding, it was mentioned that surgeons where not blinded, but it is not clear if the patient was not blinded either? Or the person in charge of assessing the outcomes, some blinding could have been done there.

Response c:

We thank the reviewer for raising this important point. Surgeons were not blinded, as noted, given the technical nature of suture placement. However, patients were not informed of their allocation, and to minimize bias, randomization results were not recorded in the medical charts but only in the case report form (CRF). Importantly, the outcome (incisional hernia) was also assessed by the operating surgeons, but because the randomization results were not documented in the medical record and were stored exclusively in the eCRF, the risk of bias during the diagnosis of incisional hernia was minimized. We have added this point to the Discussion section.

Page 18, Line 322-326

Because the surgeons were responsible for both the intervention and outcome assessment, the risk of performance and detection bias cannot be fully excluded. However, patients were not informed of their allocation, and randomization results were not documented in the medical records but stored only in the eCRF, thereby minimizing potential bias during incisional hernia diagnosis.

Comment d: I have my doubts about analysis. It is not mentioned the existence of a Data and Safety Monitoring Board, or not doing an interim analysis being the first RCT on this kind of suture.

There is a paragraph mentioning that intention to treat analysis was similar to per protocol analysis but the definition of ITT mentioned is not justified as there were approximately 20% of patients not reported, and not included in the analysis even though they were not only allocated but operated on, looking at the consort flowchart. So that introduces confusion about the definition used for ITT when there are so many loss of follow ups.

Which is part of a rigorous statistical analysis in a RCT. Some data is missing and not following CONSORT guideline, only the statistical methods to compare groups for primary and secondary outcomes is described, not including harms.

But it is missing:

• Definition of who is included in each analysis and in which group, ITT? Or Per protocol? How

Response:

We thank the reviewer for this valuable comment. In our study, the primary outcome analysis was conducted using a modified intention-to-treat (mITT) approach. Specifically, the mITT population included all randomized patients who underwent surgery and had evaluable data for incisional hernia up to 1 year post-surgery. For completeness, we have also added the definitions of both modified ITT and per-protocol analysis in the revised Methods section.

Page 10, Line 190-194

The modified intention-to-treat (mITT) population was defined as all randomized patients who underwent surgery and had evaluable data for incisional hernia up to 1 year post-surgery. The per-protocol (PP) population was defined as patients who adhered to the assigned intervention without major protocol deviations, completed scheduled follow-up, and had evaluable outcome data.

We also acknowledge that in the mITT analysis, a relatively high dropout rate and loss to follow-up were observed. These factors could have introduced potential reporting bias. We have added this point to the Discussion section to ensure transparency and to caution readers when interpreting the findings.

Page 19, Line 331-334

This study may be subject to potential reporting bias due to perioperative dropouts and loss to follow-up, as well as performance bias related to the use of reinforced sutures, additional subcutaneous sutures, and subcutaneous drains. These limitations should be taken into consideration when interpreting the findings.

• How did you handle missing data in the analysis, excluding them?

Response:

Patients who were lost to follow-up or who dropped out immediately before or during surgery could not have their primary outcome (incisional hernia occurrence) assessed; therefore, they were excluded from the analysis. This handling of missing data has been explicitly described in the first paragraph of the Results section (Page 11, Line 201-207). We acknowledge this as a limitation of our study.

• Methods for any additional analysis such as sensitivity analysis, confounders?

Response:

We thank the reviewer for this important suggestion. We acknowledge the value of sensitivity analyses in assessing the robustness of results. However, in our study the primary outcome (incisional hernia up to 1-year post-surgery) was extremely rare, with only one events in the control group and none in the barbed suture group. Because of this very low event rate, additional analyses such as worst-case versus best-case imputation of dropouts would not have materially altered the overall interpretation. Nonetheless, we carefully reviewed dropout and follow-up losses under these assumptions and confirmed that the conclusions remained unchanged. We have added this clarification to the Discussion section, while also acknowledging the limitations posed by the rarity of events.

Page 19, Line 334-342

To address concerns regarding missing data, we performed a sensitivity analysis under both best-case and worst-case assumptions. In the best-case scenario, where all dropouts and follow-up losses were assumed not to have developed incisional hernia, the results were identical to the observed outcomes (0 vs. 1 event). In the worst-case scenario, where all such patients were assumed to have developed incisional hernia, the incidence was 22.1% (19/86) in the barbed suture group and 20.5% (18/88) in the non-barbed suture group. Importantly, in both scenarios, the overall interpretation of the primary outcome remained unchanged, suggesting that our findings are robust despite the relatively high dropout and follow-up loss rates.

As shown in Table 2, there were no statistically significant differences in baseline clinical variables between the experimental and control groups. Similarly, in Table 3, surgical factors—except for suture length used and the suture length-to-wound length ratio—did not differ significantly between groups. Therefore, among the variables collected in our study, we did not identify any meaningful confounders that would require additional adjustment. Furthermore, the primary aim of this study was to compare the incidence of incisional hernia between barbed and non-barbed sutures, rather than to identify independent risk factors. For this reason, we did not perform an additional multivariable analysis.

We also acknowledge that our study did not capture several important patient-related risk factors known to influence incisional hernia development, such as metabolic syndrome, chronic obstructive pulmonary disease (COPD), malnutrition, and abdominal aortic aneurysm. This limitation has been described in the Discussion section (Page19, Line 326-328).

In our trial, a Data and Safety Monitoring Board (DSMB) was convened to oversee patient safety throughout the study period. Interim monitoring was performed, and no significant differences were observed between groups regarding wound dehiscence, infection, or incisional hernia at that stage. In addition, adverse events were systematically recorded. As described in the Results section (Page 15, Line 247-251), immediate postoperative complications during hospitalization included one case each of acute kidney injury, deep vein thrombosis, and stroke (on postoperative day 3), as well as one case of postoperative ileus that improved with conservative treatment. No complications were reported at the 4-week or 1-year follow-up visits.

Comment e. It is clear that data is not publicly available but how in who can see them

Response e:

We thank the reviewer for raising this important point regarding data availability. The data cannot be shared publicly due to the Privacy Law. However, the data are available from the Data Review Board of Seoul National University Bundang Hospital (contact: drb@snubh.org) for researchers who meet the criteria for access to confidential data. We have documented this information in the submission system and added it to the Methods section.

Page 8, Line 146-148

The data cannot be shared publicly due to the Privacy Law. However, the data are available from the Data Review Board of Seoul National University Bundang Hospital for researchers who meet the criteria for access to confidential data.

Comment f. Many CONSORT items are missing

Response f:

We thank the reviewer for pointing out this important issue. In the revised manuscript, we have updated the CONSORT flow diagram as Revised Figure 1 to more clearly present the number of patients at each stage, including eligibility assessment, randomization, allocation, follow-up, and analysis. Specifically, we have added detailed reasons for exclusion, loss to follow-up, and analysis exclusion in accordance with the CONSORT 2010 guidelines. We have also clarified which patients were included in the mITT analysis.

Comment g. Table 1 is not necessary, patients’ characteristics should have been presented by randomized groups looking for confounders

Response g:

We thank the reviewer for this valuable comment. Table 1 was included to provide an overview of the study population characteristics as a whole. Clinical and surgical factors stratified by randomized groups (experimental vs. control) are already presented in Tables 2 and 3, respectively. As shown, except for suture length used and the suture length-to-wound length ratio, no other variables demonstrated statistically significant differences between the two groups. Therefore, while Table 1 provides a useful overview of the study population, potential confounders were adequately addressed through the randomized group comparisons shown in Tables 2 and 3.

Comment h. Small sample size. Authors recognized the study is underpowered

Response h:

We thank the reviewer for this important point. Based on previous studies, the 1-year cumulative incidence of incisional hernia in the control group was estimated to be 0.13, while we assumed a lower incidence of 0.03 in the experimental group, as no existing studies had specifically evaluated barbed sutures in abdominal fascial closure. Using these assumptions, we calculated a required sample size of 87 per group (total 174 participants), setting the one-sided alpha level at 0.05 and the power at 0.8, while accounting for a 6% dropout rate (P

---

## [Decision Letter · Decision Letter 1]

4 Nov 2025

Effect of Fascial Closure Using Barbed Sutures on Incisional Hernias in Midline Laparotomy for Gynecological Diseases: A multicenter randomized controlled trial (KGOG 4001)

PONE-D-25-24135R1

Dear Dr. Kidom King,

We’re pleased to inform you that your manuscript has been judged scientifically suitable for publication and will be formally accepted for publication once it meets all outstanding technical requirements.

Kind regards,

Marco Clementi, Associate Professor

Academic Editor

PLOS ONE

Additional Editor Comments (optional):

Thank you for giving me the opportunity to evaluate such an interesting work.

In this new form the paer can be accepted for publication

Reviewers' comments:

Reviewer's Responses to Questions

**Comments to the Author**

Reviewer #1: All comments have been addressed

Reviewer #2: All comments have been addressed

2. Is the manuscript technically sound, and do the data support the conclusions?

Reviewer #1: (No Response)

Reviewer #2: Yes

3. Has the statistical analysis been performed appropriately and rigorously?

Reviewer #1: (No Response)

Reviewer #2: Yes

4. Have the authors made all data underlying the findings in their manuscript fully available?

Reviewer #1: (No Response)

Reviewer #2: Yes

5. Is the manuscript presented in an intelligible fashion and written in standard English?

Reviewer #1: (No Response)

Reviewer #2: Yes

Reviewer #1: (No Response)

Reviewer #2: Authors have adequatly addressed all previous comments. From my point of view this paper can be accepted for publication

**Do you want your identity to be public for this peer review?** For information about this choice, including consent withdrawal, please see our Privacy Policy

Reviewer #1: No

Reviewer #2: **Yes: ** Maria-Virginia Rodriguez Funes

---

## [Editor Report · Acceptance letter]

PONE-D-25-24135R1

PLOS ONE

Dear Dr. Kim,

I'm pleased to inform you that your manuscript has been deemed suitable for publication in PLOS ONE. Congratulations! Your manuscript is now being handed over to our production team.

Kind regards,

on behalf of

Dr. Marco Clementi

Academic Editor

PLOS ONE